# Removing Hidden Confounding in Recommendation: A Unified Multi-Task Learning Approach

**Haoxuan Li[1], Kunhan Wu[2], Chunyuan Zheng[3], Yanghao Xiao[4],**
**Hao Wang[5], Zhi Geng[6], Fuli Feng[7], Xiangnan He[7], Peng Wu[6,*]**

[1]Peking University      [2]Carnegie Mellon University      [3]University of California, San Diego
[4]University of Chinese Academy of Sciences      [5]Zhejiang University
[6]Beijing Technology and Business University      [7]University of Science and Technology of China

## Abstract

In recommender systems, the collected data used for training is always subject to selection bias, which poses a great challenge for unbiased learning. Previous studies proposed various debiasing methods based on observed user and item features, but ignored the effect of hidden confounding. To address this problem, recent works suggest the use of sensitivity analysis for worst-case control of the unknown true propensity, but only valid when the true propensity is near to the nominal propensity within a finite bound. In this paper, we first perform theoretical analysis to reveal the possible failure of previous approaches, including propensity-based, multi-task learning, and bi-level optimization methods, in achieving unbiased learning when hidden confounding is present. Then, we propose a unified multi-task learning approach to remove hidden confounding, which uses a few unbiased ratings to calibrate the learned nominal propensities and nominal error imputations from biased data. We conduct extensive experiments on three publicly available benchmark datasets containing a fully exposed large-scale industrial dataset, validating the effectiveness of the proposed methods in removing hidden confounding.

## 1 Introduction

Recommender systems (RS) play a key role in information retrieval by filtering out items that may be of interest to users [16, 24]. In general, the training process of recommendation uses historical user interaction data. However, a challenge in using interactions to make predictions is selection bias [3, 32, 40, 57], i.e., users always choose preferred items to interact [5, 51], resulting in a difference in the distribution between data with and without interactions [14, 45–49, 54, 56, 60], which poses a great challenge for unbiased evaluation and learning of the prediction models.

Many methods have been proposed to tackle the selection bias problem, such as the error imputation-based (EIB) [2], inverse propensity scoring (IPS) [33, 41, 42], and doubly robust (DR) methods [7, 11, 27–29, 52]. Based on these, recent works further incorporate multi-task learning to alleviate the data sparsity issue, such as entire space multi-task model (ESMM) [34], multi-task IPS and multi-task DR (Multi-IPS and Multi-DR) [59], and entire space counterfactual multi-task models (ESCM$^2$) [44].

However, these studies only consider the selection bias induced by measured confounders, but ignore the presence of hidden (or unmeasured) confounders, which considerably influences the application of these advanced methods in real-world recommendation scenarios. Hidden confounders are ubiquitous and inevitable in RS due to information limitations (e.g., friend useful suggestions) or privacy restrictions (e.g., user salary) [8, 31, 35, 50]. In this paper, we perform theoretical analysis

---

*Corresponding author: pengwu@btbu.edu.cn.

37th Conference on Neural Information Processing Systems (NeurIPS 2023).

for the existing propensity-based and multi-task learning methods, showing that all of them will lead to biased evaluation and learning in the presence of hidden confounding.

To remove hidden confounding, previous causal inference literature suggested the use of instrumental variables [12] or front door adjustment [36]. However, in real-world application scenarios, these approaches require strong assumptions that are difficult to verify in practice. On the other hand, a recent recommendation study proposed robust deconfounder (RD) to adopt sensitivity analysis using a min-max optimization for worst-case control of the unknown true propensity [8]. Nevertheless, they assume the true propensity is near to the nominal one within a bound, which is decided by the strength of the unmeasured confounder, posing another challenge in case these assumptions are violated.

In contrast, unbiased datasets are regarded as the gold standard for unbiased evaluations and can be collected from A/B tests or randomized controlled trials [10, 13, 21, 55]. This provides an alternative solution to remove hidden confounding [15, 20, 26, 58]. Nevertheless, training recommendation models directly on unbiased datasets can suffer from severe overfitting due to the limited sample size restricted by the collection cost of random exposure [4]. Despite few works leverage the unbiased data to combat selection bias [4, 53] by adopting the bi-level optimization, this paper shows that they still lead to biased learning, due to the biased hypothesis space of the prediction model.

To tackle the above problems, we propose a unified multi-task learning approach to remove hidden confounding by leveraging a few unbiased ratings. Interestingly, we show that the unbiased data can help calibrate the learned nominal propensities and nominal error imputations, which differs from RD using sensitivity analysis for nominal propensities only [8]. Specifically, the proposed multi-task learning builds residual networks for learned propensities and imputed errors from biased data, respectively. Next, a consistency loss of IPS (or DR) estimation on the biased dataset and empirical average prediction errors on the unbiased dataset are developed to help the training of the two residual networks, for calibrating the learned nominal propensities and nominal error imputations, respectively. The prediction model is then trained with the calibrated IPS (or calibrated DR) loss and unbiased data to achieve unbiased learning in the presence of hidden confounding.

The main contributions of this paper are summarized as follows:

- We theoretically reveal limitations of the existing multi-task learning and bi-level optimization methods for achieving the unbiasedness in the presence of hidden confounding.

- We provide a unified multi-task learning approach to remove hidden confounding by combining a few unbiased ratings, in which the learned nominal propensities and nominal error imputations can be calibrated by the residual networks with the proposed consistency loss.

- We conduct extensive experiments on three publicly available benchmark datasets, including a large-scale industrial dataset, to validate the effectiveness of the proposed methods.

## 2 Problem Setup

We formulate the selection bias in RS using post-click conversion rate (pCVR) prediction task for illustration purpose, which can be naturally generalized to other recommendation tasks with explicit feedback, e.g., rating prediction. Suppose that the entire space has $m$ users and $n$ items, let $\mathcal{D}$ be the set of all user-item pairs. Denote $\mathbf{R} \in \{0, 1\}^{m \times n}$ as the true post-click conversion label matrix of user-item pairs, where each entry $r_{u,i}$ indicates whether a conversion occurs after user $u$ clicks on item $i$. Let $x_{u,i}$ be the feature of user-item pair $(u, i)$, and $\hat{\mathbf{R}} \in \mathbb{R}^{m \times n}$ be the prediction matrix for pCVR, where $\hat{r}_{u,i} = f(x_{u,i}, \theta_{\text{CVR}}) \in [0, 1]$ is the predicted pCVR obtained by a model $f$ with parameter $\theta_{\text{CVR}}$. In RS, users always select the preferred items to click on, leading to a significant difference in the distribution between clicked and unclicked events thus causing selection bias.

If $\mathbf{R}$ is fully observed, then a pCVR model $f(x_{u,i}; \theta_{\text{CVR}})$ can be trained by minimizing the ideal loss

$$\mathcal{L}_{\text{ideal}}(\theta_{\text{CVR}}) = \frac{1}{|\mathcal{D}|} \sum_{(u,i) \in \mathcal{D}} \delta_{u,i},$$

where $\delta_{u,i} \triangleq \delta(r_{u,i}, \hat{r}_{u,i})$ and $\delta(\cdot, \cdot)$ is a pre-specified loss, e.g., the cross-entropy loss, $\delta(r_{u,i}, \hat{r}_{u,i}) = -r_{u,i} \log \hat{r}_{u,i} - (1 - r_{u,i}) \log (1 - \hat{r}_{u,i})$. However, the post-click conversion feedback of a user-item pair $(u, i)$ can be observed only when user $u$ clicks on item $i$, making the ideal loss not computable. Let $o_{u,i}$ be the indicator of user $u$ clicking on item $i$, and $\mathcal{B} = \{(u, i) \mid (u, i) \in \mathcal{D}, o_{u,i} = 1\}$ be

the set of clicked events, where $\mathcal{B}$ means that the clicked events is a biased sample of the entire space $\mathcal{D}$. In this paper, we further consider the presence of hidden confounding. Without loss of generality, we assume that all confounders consist of a measured part $x_{u,i}$ and a hidden (unmeasured) part $h_{u,i}$, where the latter arises from issues such as information limitations (e.g., friend suggestions) and privacy restrictions (e.g., user salary), which cannot be observed explicitly and used for training.

# 3 Previous Methods Lead to Biased Learning under Hidden Confounding

## 3.1 Multi-Task Learning

The pCVR prediction task is closely related to click-through rate (CTR) and post-view click-through & conversion rate (CTCVR) prediction tasks, as the formula CTCVR = CTR $*$ pCVR holds, where the CTR is $p_{u,i} \triangleq \mathbb{P}(o_{u,i} = 1 | x_{u,i})$, also known as propensity score in the causal inference literature, represents the probability of a user $u$ clicking on an item $i$. The CTCVR is $\mathbb{P}(r_{u,i} = 1, o_{u,i} = 1 | x_{u,i})$, means the probability that item $i$ is clicked and converted by user $u$.

Let $\hat{p}_{u,i} \triangleq \hat{p}_{u,i}(x_{u,i}, \theta_{\text{CTR}})$ be CTR prediction model with parameter $\theta_{\text{CTR}}$. The ESMM method [34] learns pCVR by joint-training CTR and CTCVR losses

$$\mathcal{L}_{\text{CTR}}(\theta_{\text{CTR}}) = \frac{1}{|\mathcal{D}|} \sum_{(u,i) \in \mathcal{D}} \delta\left(o_{u,i}, \hat{p}_{u,i}\right), \quad \mathcal{L}_{\text{CTCVR}}(\theta_{\text{CTR}}, \theta_{\text{CVR}}) = \frac{1}{|\mathcal{D}|} \sum_{(u,i) \in \mathcal{D}} \delta\left(o_{u,i} r_{u,i}, \hat{p}_{u,i} \hat{r}_{u,i}\right).$$

However, the ESMM loss $\mathcal{L}_{\text{CTR}}(\theta_{\text{CTR}}) + \mathcal{L}_{\text{CTCVR}}(\theta_{\text{CTR}}, \theta_{\text{CVR}})$ is a biased estimator of the ideal loss [59]. To achieve unbiased learning, the MTL-IPS and MTL-DR methods [59] use the losses

$$\mathcal{L}_{\text{IPS}}(\theta_{\text{CTR}}, \theta_{\text{CVR}}) = \frac{1}{|\mathcal{D}|} \sum_{(u,i) \in \mathcal{D}} \frac{o_{u,i} \delta_{u,i}}{\hat{p}_{u,i}}, \ \mathcal{L}_{\text{DR}}(\theta_{\text{CTR}}, \theta_{\text{CVR}}, \theta_{\text{IMP}}) = \frac{1}{|\mathcal{D}|} \sum_{(u,i) \in \mathcal{D}} \left[\hat{\delta}_{u,i} + \frac{o_{u,i}(\delta_{u,i} - \hat{\delta}_{u,i})}{\hat{p}_{u,i}}\right],$$

where $\hat{\delta}_{u,i} \triangleq \hat{\delta}_{u,i}(x_{u,i}, \theta_{\text{IMP}})$ is the imputation model that predicts $\delta_{u,i}$ using $x_{u,i}$, i.e., it estimates $g_{u,i} \triangleq \mathbb{E}[\delta_{u,i} | x_{u,i}]$. Without hidden confounders, $\mathcal{L}_{\text{IPS}}$ is an unbiased estimator of the ideal loss when the learned propensities are accurate, i.e., $\hat{p}_{u,i} = p_{u,i}$ [41, 42], and $\mathcal{L}_{\text{DR}}$ is unbiased if either $\hat{p}_{u,i} = p_{u,i}$ or $\hat{\delta}_{u,i} = g_{u,i}$ [39, 52]. However, both IPS and DR are biased under hidden confounders.

**Lemma 1** (Theorem 3.1 in [8]). *In the presence of hidden confounders $h_{u,i}$, both $\mathcal{L}_{IPS}$ and $\mathcal{L}_{DR}$ are biased estimators of the ideal loss, even if $\hat{p}_{u,i} = p_{u,i}$ and $\hat{\delta}_{u,i} = g_{u,i}$.*

The above result also holds in the follow-up DR studies [23, 25, 43], and can be naturally extended to conclude that the MTL-IPS and MTL-DR [59] are biased under hidden confounding. Recently, ESCM$^2$-IPS and ESCM$^2$-DR showed state-of-the-art performance in pCVR prediction by incorporating the ESMM loss $\mathcal{L}_{\text{CTCVR}}$ as the global risk of CTCVR, and MTL-IPS and MTL-DR losses $\mathcal{L}_{\text{CVR}}^{\mathcal{B}}$ as counterfactual risk of pCVR [44]. Formally, the ESCM$^2$ loss is

$$\mathcal{L}_{\text{ESCM}^2} = \mathcal{L}_{\text{CTR}} + \lambda_1 \mathcal{L}_{\text{IMP}} + \lambda_2 \mathcal{L}_{\text{CVR}}^{\mathcal{B}} + \lambda_3 \mathcal{L}_{\text{CTCVR}},$$

where $\lambda_t$ for $t = 1, 2, 3$ are hyper-parameters, $\mathcal{L}_{\text{CVR}}^{\mathcal{B}}$ is either $\mathcal{L}_{\text{IPS}}$ or $\mathcal{L}_{\text{DR}}$. The imputation loss is

$$\mathcal{L}_{\text{IMP}}(\theta_{\text{CTR}}, \theta_{\text{CVR}}, \theta_{\text{IMP}}) = \frac{1}{|\mathcal{B}|} \sum_{(u,i) \in \mathcal{B}} \frac{(\delta_{u,i} - \hat{\delta}_{u,i})^2}{\hat{p}_{u,i}}.$$

However, since both $\mathcal{L}_{\text{CTCVR}}$ and $\mathcal{L}_{\text{CVR}}^{\mathcal{B}}$ are biased under hidden confounding, ESCM$^2$ is also biased.

## 3.2 Debiasing with a Few Unbiased Ratings

Instead of only using the biased dataset $\mathcal{B}$, many methods are proposed to improve the debiasing performance by combining a small unbiased dataset $\mathcal{U}$ and a large biased dataset $\mathcal{B}$, such as bi-level optimization approaches, including learning to debias (LTD) [53] and AutoDebias [4], causal embedding method (CausE) [1], knowledge distillation framework for counterfactual recommendation via uniform data (KDCRec) [30, 32], and causal balancing methods [26].

Specifically, learning to debias (LTD) [53] and AutoDebias [4] adopt bi-level optimization [17, 37] to learn a CTR model $\hat{p}_{u,i}(x_{u,i}, \theta_{\text{CTR}})$ such that the pCVR prediction model $\hat{r}_{u,i} = f(x_{u,i}, \theta_{\text{CVR}})$ performs well on the small unbiased dataset $\mathcal{U}$. Formally, the bi-level optimization in LTD is

$$\theta_{\text{CTR}}^* = \arg\min_{\theta_{\text{CTR}}} \mathcal{L}_{\text{CVR}}^{\mathcal{U}}(\theta_{\text{CVR}}^*(\theta_{\text{CTR}})),$$

$$\text{s.t. } \theta_{\text{CVR}}^*(\theta_{\text{CTR}}) = \arg\min_{\theta_{\text{CVR}}} \mathcal{L}_{\text{CVR}}^{\mathcal{B}}(\theta_{\text{CTR}}, \theta_{\text{CVR}}),$$

where the upper loss is as an average of prediction errors on the unbiased dataset

$$\mathcal{L}_{\text{CVR}}^{\mathcal{U}}(\theta_{\text{CVR}}^*(\theta_{\text{CTR}})) = \frac{1}{|\mathcal{U}|} \sum_{(u,i)\in\mathcal{U}} \delta\left(r_{u,i}, \hat{r}_{u,i}(\theta_{\text{CVR}}^*(\theta_{\text{CTR}}))\right),$$

and AutoDebias further develop a solution for universal debiasing. However, despite the use of unbiased ratings, we show that they still lead to biased estimates under hidden confounding.

**Proposition 1.** *In the presence of hidden confounders, both LTD and AutoDebias are biased.*

*Proof.* Let $\mathcal{H}_{\text{CTR}} = \{\hat{p}_{u,i}(x_{u,i}, \theta_{\text{CTR}}) : \theta_{\text{CTR}} \in \Theta_{\text{CTR}}\}$ be the hypothesis space of propensity model. Then $\theta_{\text{CTR}}^*$ defined above is the parameter in $\Theta_{\text{CTR}}$ such that the pCVR prediction model performs optimally on $\mathcal{U}$. The bi-level optimization takes $\mathcal{L}_{\text{CVR}}^{\mathcal{B}}$ to estimate the ideal loss using a selected propensity $\hat{p}_{u,i}(x_{u,i}, \theta_{\text{CTR}}^*) \in \mathcal{H}_{\text{CTR}}$, however, $\mathcal{L}_{\text{CVR}}^{\mathcal{B}}$ could be a biased estimator of the ideal loss for all $\hat{p}_{u,i} \in \mathcal{H}_{\text{CTR}}$ (due to $\hat{p}_{u,i}$ could deviates from the true one by an arbitrary distance, as formally stated in Theorem 2), therefore both LTD and AutoDebias using bi-level optimization are biased. □

Proposition 1 formally reveals the limitations of directly adopting bi-level optimization for addressing hidden confounding. Both LTD and AutoDebias essentially use unbiased data for *model selection* among all possible IPS or DR losses, but in fact, without *correcting* the IPS or DR estimators themselves for predicting the ideal loss, those IPS or DR-based methods will not be able to tackle unobserved confounding, because of the intrinsic biasedness of the estimators to the ideal loss.

An alternative class of methods that use unbiased data for tuning is the causal embedding method (CausE) [1] and KDCRec [30, 32]. Specifically, they both consider building a connection between a model trained with biased data and another model trained with unbiased data. CausE [1] designs an alignment term as the pairwise difference between the parameters of the two models, which is then included in the object function to be minimized. KDCRec [30, 32] proposes a general knowledge distillation framework for counterfactual recommendation via uniform data, including label-based, feature-based, sample-based, and model structure-based distillations. The above methods empirically shown impressive performance by distilling the shared information on both biased and unbiased data. However, theoretical guarantees under hidden confounding are lacking, and it would be interesting to investigate the conditions under which these methods can achieve unbiased learning.

### 3.3 Mitigating Hidden Confounding with Sensitivity Analysis

To tackle the problem of hidden confounding, robust deconfounder (RD) [8] proposes to adopt sensitivity analysis from the causal inference literature [6, 38] to minimize the worst-case prediction loss. Formally, the unknown **true propensity** $\bar{p}_{u,i} \triangleq \mathbb{P}(o_{u,i} = 1 | x_{u,i}, h_{u,i})$ is assumed to near the **nominal propensity** $p_{u,i} = \mathbb{P}(o_{u,i} = 1 | x_{u,i})$ within a bound that

$$\frac{1}{\Gamma} \leq \frac{(1-p_{u,i})\bar{p}_{u,i}}{p_{u,i}(1-\bar{p}_{u,i})} \leq \Gamma \implies 1 + (1/p_{u,i} - 1)/\Gamma \triangleq a_{u,i} \leq w_{u,i} \leq b_{u,i} \triangleq 1 + (1/p_{u,i} - 1)\Gamma,$$

where hyper-parameter $\Gamma$ corresponds to the strength of unmeasured confounding, $w_{u,i} = 1/\bar{p}_{u,i}$ is the inverse of the true propensity, and $a_{u,i}$ and $b_{u,i}$ are the lower and upper bounds of $w_{u,i}$, respectively. Let $\mathcal{W} = [\hat{a}_{1,1}, \hat{b}_{1,1}] \times \cdots \times [\hat{a}_{m,n}, \hat{b}_{m,n}]$ be the possible inverse propensities on all user-item pairs, where $\hat{a}_{u,i}$ and $\hat{b}_{u,i}$ are the estimates of $a_{u,i}$ and $b_{u,i}$, respectively, then RD-IPS method trains the prediction model by minimizing the worst-case IPS loss

$$\mathcal{L}_{\text{RD-IPS}}(\theta_{\text{CTR}}, \theta_{\text{CVR}}) = \max_{W \in \mathcal{W}} \frac{1}{|\mathcal{D}|} \sum_{(u,i)\in\mathcal{D}} o_{u,i} \delta_{u,i} w_{u,i},$$

and the RD-DR method can be developed by a similar argument controlling the worst-case DR loss.

To summarize, the RD methods first obtains the bounds of true propensities around the nominal propensities, then minimize the upper bound of the IPS (or DR) loss to control the worst-case caused by unmeasured confounders. However, on the one hand, it is not clear how to set $\Gamma$ correctly since both the true propensities and the strength of hidden confounding are unknown. On the other hand, the effectiveness of sensitivity analysis for controlling the hidden confounding requires that the true propensity is around the nominal propensity for all user-item pairs, but such (strong) assumptions cannot be verified from the data and raises another concern in case the assumptions are violated.

## 4 Debiasing Residual Networks under Hidden Confounding

### 4.1 Methodology Overview

Different from the previous methods that use sensitivity analysis to control the worst-case IPS or DR loss caused by hidden confounding, we propose a unified multi-task learning approach with residual networks as in Figure 1, with the motivation of using a small unbiased data to calibrate the learned propensities and imputed errors in IPS or DR loss for training the unbiased prediction model.

Following causal inference literature [18, 19], we define the **true propensity** and **true imputation** as $\bar{p}_{u,i} \triangleq \mathbb{P}(o_{u,i} = 1 | x_{u,i}, h_{u,i})$, $\bar{g}_{u,i} \triangleq \mathbb{E}(\delta_{u,i} | x_{u,i}, h_{u,i})$, both of them are functions of $(x_{u,i}, h_{u,i})$, with $\tilde{p}_{u,i}$ and $\tilde{\delta}_{u,i}$ as their estimates. To distinguish, we call $p_{u,i} = \mathbb{P}(o_{u,i} = 1 | x_{u,i})$ and $g_{u,i} = \mathbb{E}[\delta_{u,i} | x_{u,i}]$ the **nominal propensity** and **nominal imputation**, with $\hat{p}_{u,i}$ and $\hat{\delta}_{u,i}$ as their estimates.

Next, we show the necessity of calibrations on both $\hat{p}_{u,i}$ and $\hat{\delta}_{u,i}$ estimated from the baised data $\mathcal{B}$.

**Theorem 2** (Necessity of Calibration). *Suppose the partial derivative of $\bar{p}_{u,i}$ and $\bar{g}_{u,i}$ with respect to hidden confounders $h_{u,i}$ are not always equal to 0, and $\hat{p}_{u,i}$ and $\hat{\delta}_{u,i}$ are consistent estimators of $p_{u,i}$ and $g_{u,i}$, then there exists $\eta > 0$, such that*

$$\lim_{|\mathcal{D}| \to \infty} \mathbb{P}(|\hat{p}_{u,i} - \bar{p}_{u,i}| > \eta) > 0, \quad \lim_{|\mathcal{D}| \to \infty} \mathbb{P}(|\hat{\delta}_{u,i} - \bar{g}_{u,i}| > \eta) > 0.$$

*Proof.* Given the partial derivative of $\bar{p}_{u,i}$ with respect to hidden confounders $h_{u,i}$ is not always equal to 0, that is, $h_{u,i}$ has a non-zero effect on $o_{u,i}$, so we have $\bar{p}_{u,i} \neq p_{u,i}$ according to their definitions. Thus, for some $\epsilon > 0$, there exist positive constants $\delta_\epsilon, N_1(\epsilon) > 0$, such that for all $|\mathcal{D}| > N_1(\epsilon)$,

$$\mathbb{P}(|\bar{p}_{u,i} - p_{u,i}| > \epsilon) > \delta_\epsilon > 0.$$

Since $\hat{p}_{u,i}$ is a consistent estimator of $p_{u,i}$, there exists some $N_2(\epsilon) > 0$, such that for all $|\mathcal{D}| > N_2(\epsilon)$,

$$\mathbb{P}(|\hat{p}_{u,i} - p_{u,i}| \geq \epsilon/2) < \frac{\delta_\epsilon}{4}.$$

Thus, if $|\mathcal{D}| > \max\{N_1(\epsilon), N_2(\epsilon)\}$, we have

$$\begin{aligned}
& \mathbb{P}(|\bar{p}_{u,i} - p_{u,i}| > \epsilon, |\hat{p}_{u,i} - p_{u,i}| < \epsilon/2) \\
&= \mathbb{P}(|\bar{p}_{u,i} - p_{u,i}| > \epsilon) - \mathbb{P}(|\bar{p}_{u,i} - p_{u,i}| > \epsilon, |\hat{p}_{u,i} - p_{u,i}| \geq \frac{\epsilon}{2}) \\
&> \delta_\epsilon - \delta_\epsilon/4 = 3\delta_\epsilon/4.
\end{aligned}$$

Let $\eta = \epsilon/2$ and note that $\{|\bar{p}_{u,i} - p_{u,i}| > \epsilon, |\hat{p}_{u,i} - p_{u,i}| < \epsilon/2\} \subset \{|\hat{p}_{u,i} - \bar{p}_{u,i}| > \eta\}$, we have

$$\mathbb{P}(|\hat{p}_{u,i} - \bar{p}_{u,i}| > \eta) \geq \mathbb{P}(|\bar{p}_{u,i} - p_{u,i}| > \epsilon, |\hat{p}_{u,i} - p_{u,i}| < \epsilon/2) > 3\delta_\epsilon/4 > 0,$$

which leads to $\lim_{|\mathcal{D}| \to \infty} \mathbb{P}(|\hat{p}_{u,i} - \bar{p}_{u,i}| > \eta) > 0$. Similarly, it can be shown that $\lim_{|\mathcal{D}| \to \infty} \mathbb{P}(|\hat{\delta}_{u,i} - \bar{g}_{u,i}| > \eta) > 0$. □

Theorem 2 shows that in the presence of hidden confounding, the estimated nominal propensities and nominal imputed errors deviate from the true one, even with the infinite sample size. To address this problem, as shown in Figure 1, we propose a novel consistency loss that utilizes unbiased data to calibrate the learned nominal propensities and imputed errors from the biased data.

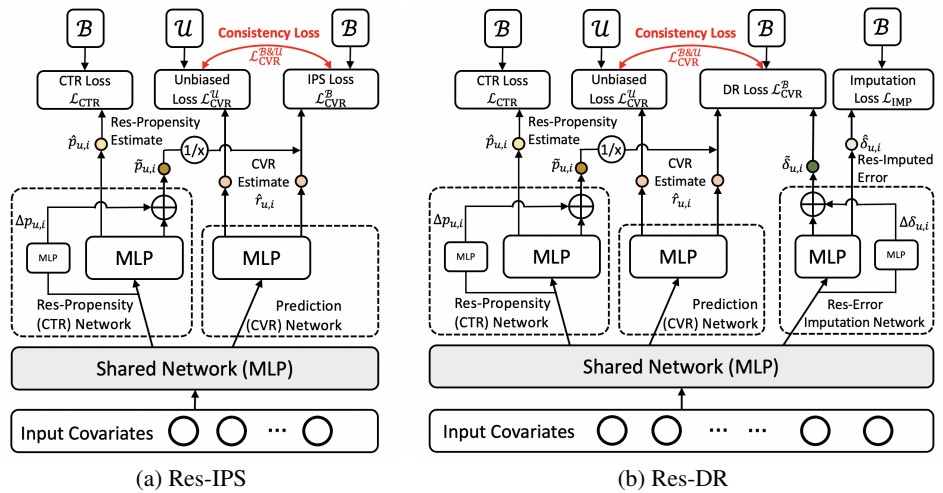

Figure 1: Proposed debiasing residual networks for removing hidden confounding.

Specifically, we define the **calibrated propensity model** $\tilde{p} = \tilde{p}(\theta_{\text{CTR}}, \phi_{\text{CTR}})$ and the **calibrated imputation model** $\tilde{\delta} = \tilde{\delta}(\theta_{\text{IMP}}, \phi_{\text{IMP}})$ as follows

$$\tilde{p}_{u,i} = \sigma\left(\sigma^{-1}(\hat{p}_{u,i}(\theta_{\text{CTR}})) + \sigma^{-1}(\Delta p_{u,i}(\phi_{\text{CTR}}))\right),$$

$$\tilde{\delta}_{u,i} = \sigma\left(\sigma^{-1}(\hat{\delta}_{u,i}(\theta_{\text{IMP}})) + \sigma^{-1}(\Delta\delta_{u,i}(\phi_{\text{IMP}}))\right),$$

where $\sigma$ is the sigmoid function, and the transformations are designed for numerical stability, e.g., to control range from 0 to 1. Compared with $\hat{p}_{u,i}$ and $\hat{\delta}_{u,i}$ adopted in the previous multi-task learning approaches [34, 44, 59], the residual terms $\Delta p_{u,i}$ and $\Delta\delta_{u,i}$ are further added to $\tilde{p}_{u,i}$ and $\tilde{\delta}_{u,i}$ to capture the effect of hidden confounding. The loss function of the proposed approach is defined as

$$\mathcal{L}_{\text{Res}} = \underbrace{\mathcal{L}_{\text{CTR}}(\hat{p}) + \alpha \cdot \mathcal{L}_{\text{IMP}}(\hat{\delta})}_{\text{Initialization using biased data}} + \beta \cdot \underbrace{\left(\mathcal{L}_{\text{CVR}}^{\mathcal{B}}(\tilde{p}, \tilde{\delta}) + \mathcal{L}_{\text{CTCVR}}^{\mathcal{B}}(\tilde{p}) + \mathcal{L}_{\text{CVR}}^{\mathcal{U}}\right)}_{\text{Prediction model training with calibrated losses}} + \gamma \cdot \underbrace{\mathcal{L}_{\text{CVR}}^{\mathcal{B}\&\mathcal{U}}(\hat{p}, \tilde{p}, \hat{\delta}, \tilde{\delta})}_{\text{Calibration on } \hat{p} \text{ and } \hat{\delta}},$$

where $\alpha$, $\beta$ and $\gamma$ are hyper-parameters for trade-off. The following states for each loss.

## 4.2 Nominal Propensities and Imputations Initialization

Similar to the previous IPS and DR methods [39, 41, 42, 52], as well as the multi-task learning approaches [34, 44, 59], we use the losses $\mathcal{L}_{\text{CTR}}$ and $\mathcal{L}_{\text{IMP}}$ in Section 3.1 for training the estimated nominal propensities $\hat{p}_{u,i}$ and nominal imputed errors $\hat{\delta}_{u,i}$ from the biased dataset. However, since $\hat{p}_{u,i}$ and $\hat{\delta}_{u,i}$ can only capture the effect of *measured* confounding, we do not directly use them for training the prediction model under *hidden* confounding. Instead, we train the prediction model using the calibrated propensities and imputed errors as illustrated in Section 4.3.

## 4.3 Debiased Prediction Model Training

**Training on Biased Data.** To train an unbiased prediction model, the losses need to be unbiased with respect to the ideal loss, thus the direct use of nominal $\hat{p}_{u,i}$ and $\hat{\delta}_{u,i}$ under hidden confounding would lead to biased predictions. In contrast to previous studies, we propose to use both the initialized $\hat{p}_{u,i}$ and $\hat{\delta}_{u,i}$ in Section 4.2, and the corresponding residual terms $\Delta p_{u,i}$ and $\Delta\delta_{u,i}$ (see Section 4.4 for more details) using IPS (or DR, CTCVR loss) to achieve unbiased prediction model learning.

Specifically, using the calibrated propensity model, the proposed residual-IPS (Res-IPS) loss is

$$\mathcal{L}_{\text{IPS}}^{\mathcal{B}}(\theta_{\text{CVR}} \mid \hat{p}_{u,i}(\theta_{\text{CTR}}), \Delta p_{u,i}(\phi_{\text{CTR}})) = \frac{1}{|\mathcal{D}|} \sum_{(u,i)\in\mathcal{D}} \frac{o_{u,i} \cdot \delta_{u,i}}{\tilde{p}_{u,i}}.$$

Similarly, using both the calibrated propensity model and the calibrated imputation model, the proposed residual-DR (Res-DR) loss is

$$\mathcal{L}^{\mathcal{B}}_{\text{DR}}(\theta_{\text{CVR}} \mid \hat{p}_{u,i}(\theta_{\text{CTR}}), \Delta p_{u,i}(\phi_{\text{CTR}}), \hat{\delta}_{u,i}(\theta_{\text{IMP}}), \Delta \delta_{u,i}(\phi_{\text{IMP}})) = \frac{1}{|\mathcal{D}|} \sum_{(u,i) \in \mathcal{D}} \left[ \tilde{\delta}_{u,i} + \frac{o_{u,i}(\delta_{u,i} - \tilde{\delta}_{u,i})}{\tilde{p}_{u,i}} \right].$$

Although CTCVR loss is not unbiased to the ideal loss [59], as previous studies have shown, using both IPS (or DR) loss and CTCVR loss empirically can lead to better debiasing performance by mitigating the data sparsity issue [44]. Similarly, using the calibrated propensity model, the proposed calibrated CTCVR loss is

$$\mathcal{L}_{\text{CTCVR}}(\theta_{\text{CVR}} \mid \hat{p}_{u,i}(\theta_{\text{CTR}}), \Delta p_{u,i}(\phi_{\text{CTR}})) = \frac{1}{|\mathcal{D}|} \sum_{(u,i) \in \mathcal{D}} \delta\left(o_{u,i} r_{u,i}, \tilde{p}_{u,i} \hat{r}_{u,i}\right).$$

Empirically, one can choose from the three calibrated losses for training a debiased prediction model.

**Training on Unbiased Data.** Since the unbiased data do not encounter any confounding problems, it provides a golden standard to evaluate and train the pCVR prediction model. The unbiased loss is

$$\mathcal{L}^{\mathcal{U}}_{\text{CVR}}(\theta_{\text{CVR}}) = \frac{1}{|\mathcal{U}|} \sum_{(u,i) \in \mathcal{U}} \delta\left(r_{u,i}, \hat{r}_{u,i}\left(\theta_{\text{CVR}}\right)\right).$$

Different from the previous studies [4, 53], in which $\mathcal{L}^{\mathcal{U}}_{\text{CVR}}$ is used to select the optimal propensity or imputation models, our approach is not necessary for training the prediction model using the unbiased loss (since one can use the aforementioned calibrated losses). In addition, the direct use of the unbiased loss can lead to severe overfitting, which once again demonstrates the importance of calibrating the propensity and imputation models in the IPS or DR loss.

### 4.4 Residual Networks Training

Collected through a carefully designed experiment, the unbiased data can be regarded as a representative sample of the entire space [53]. Thus, we always have $\mathcal{L}^{\mathcal{U}}_{\text{CVR}} \approx \mathcal{L}_{\text{ideal}}(\theta_{\text{CVR}})$, regardless of hidden confounding in biased data, which motivates us to propose a consistency loss

$$\mathcal{L}^{\mathcal{B}\&\mathcal{U}}_{\text{CVR}}(\theta_{\text{CTR}}, \phi_{\text{CTR}}, \theta_{\text{CVR}}, \theta_{\text{IMP}}, \phi_{\text{IMP}}) = \delta(\mathcal{L}^{\mathcal{B}}_{\text{CVR}}, \mathcal{L}^{\mathcal{U}}_{\text{CVR}}),$$

which measures the discrepancy between $\mathcal{L}^{\mathcal{B}}_{\text{CVR}}$ and $\mathcal{L}^{\mathcal{U}}_{\text{CVR}}$, which equivalently provides us with *an optimization direction for removing hidden confounding.*

**Proposition 3.** *If $\mathcal{L}^{\mathcal{B}\&\mathcal{U}}_{CVR} = 0$, then $\mathcal{L}^{\mathcal{B}}_{CVR}$ is an unbiased estimator of the ideal loss, regardless of whether hidden confounders exist or not.*

Essentially, the consistency loss uses unbiased data to calibrate the debiasing loss $\mathcal{L}^{\mathcal{B}}_{\text{CVR}}$ based on the biased data, thereby guaranteeing the debiasing ability of the proposed methods in the presence of hidden confounding. As discussed in Sections 4.1–4.3, we utilize the two residual terms $\Delta p_{u,i}$ and $\Delta \delta_{u,i}$ to capture the effect of hidden confounding, which are trained by mimimizing $\mathcal{L}^{\mathcal{B}\&\mathcal{U}}_{\text{CVR}}$.

## 5 Real-World Experiments

**Dataset and Pre-processing.** Following the previous studies [4, 42, 52], we use three real-world datasets: COAT[2], YAHOO! R3[3], and KUAIREC[4] [9], for evaluating the debiasing performance of the proposed methods, where KUAIREC is a public large-scale industrial dataset. COAT contains 6,960 biased ratings from 290 users to 300 items, where each user picks 24 items to rate based on their personal preferences. Meanwhile, it also contains 4,640 unbiased ratings, where each user is asked to rate 16 randomly selected items. YAHOO! R3 contains 311,704 biased ratings and 54,000 unbiased ratings, where the unbiased ratings are from the first 5,400 users for 10 random selected items. We binarize ratings less than four to 0 and other ratings to 1 for the above two five-scale

---

[2]https://www.cs.cornell.edu/~schnabts/mnar/
[3]http://webscope.sandbox.yahoo.com/
[4]https://github.com/chongminggao/KuaiRec

Table 1: Performance in terms of AUC, NDCG@K, and Recall@K on the unbiased dataset of COAT, YAHOO! R3 and KUAIREC. The best two results are bolded, and the best baseline is underlined.

| Method | COAT AUC | N@5 | R@5 | YAHOO! R3 AUC | N@5 | R@5 | KUAIREC AUC | N@50 | R@50 |
|---|---|---|---|---|---|---|---|---|---|
| MF [22] (Bias) | 0.747 | 0.500 | 0.546 | 0.721 | 0.553 | 0.716 | 0.820 | 0.561 | 0.816 |
| MF [22] (Uniform) | 0.580 | 0.363 | 0.386 | 0.574 | 0.455 | 0.611 | 0.664 | 0.491 | 0.816 |
| MF [22] (Combine) | 0.751 | 0.504 | 0.546 | 0.724 | 0.558 | 0.717 | 0.822 | 0.566 | 0.812 |
| CausE [1] | 0.763 | 0.512 | 0.575 | 0.730 | 0.555 | 0.736 | 0.819 | 0.581 | 0.856 |
| ESMM [34] | 0.745 | 0.506 | 0.525 | 0.708 | 0.545 | 0.693 | 0.823 | 0.563 | 0.852 |
| KD-Label [30] | 0.760 | 0.509 | 0.562 | 0.726 | 0.583 | 0.752 | 0.815 | 0.570 | 0.858 |
| AutoDebias [4] | 0.762 | 0.540 | 0.580 | 0.735 | 0.632 | **0.785** | 0.818 | 0.584 | 0.866 |
| KD-Feature [30] | 0.766 | 0.522 | 0.584 | 0.717 | 0.557 | 0.736 | 0.809 | 0.588 | 0.873 |
| IPS [42] | 0.761 | 0.513 | 0.566 | 0.722 | 0.555 | 0.733 | 0.826 | 0.574 | 0.849 |
| Multi-IPS [59] | 0.758 | 0.514 | 0.531 | 0.719 | 0.546 | 0.710 | 0.810 | 0.554 | 0.875 |
| ESCM$^2$-IPS [44] | 0.757 | 0.514 | 0.558 | 0.729 | 0.559 | 0.714 | 0.815 | 0.577 | 0.860 |
| RD-IPS [8] | 0.764 | 0.514 | 0.566 | 0.730 | 0.571 | 0.735 | 0.832 | 0.585 | 0.873 |
| BRD-IPS [8] | 0.763 | 0.511 | 0.564 | 0.735 | 0.582 | 0.743 | 0.834 | 0.588 | 0.877 |
| Res-IPS (ours) | **0.777** | **0.575** | **0.601** | **0.759** | **0.639** | **0.785** | **0.849** | **0.601** | **0.885** |
| DR [52] | 0.766 | 0.525 | 0.552 | 0.725 | 0.553 | 0.727 | 0.824 | 0.567 | 0.838 |
| Multi-DR [59] | 0.759 | 0.527 | 0.565 | 0.719 | 0.553 | 0.712 | 0.829 | 0.562 | 0.859 |
| ESCM$^2$-DR [44] | 0.760 | 0.553 | 0.568 | 0.715 | 0.566 | 0.722 | 0.827 | 0.569 | 0.830 |
| RD-DR [8] | 0.768 | 0.539 | 0.571 | 0.732 | 0.569 | 0.738 | 0.833 | 0.585 | **0.884** |
| BRD-DR [8] | 0.770 | 0.546 | 0.577 | 0.735 | 0.576 | 0.737 | 0.831 | 0.585 | 0.883 |
| Res-DR (ours) | **0.793** | **0.588** | **0.607** | **0.750** | **0.654** | **0.803** | **0.854** | **0.595** | 0.860 |

Table 2: Effects of varying unbiased data ratio on KUAIREC in terms of AUC and NDCG@50. The best two results are bolded, and the best baseline is underlined.

| Unbiased data ratio | AUC ↑ 2% | 4% | 6% | 8% | 10% | NDCG@50 ↑ 2% | 4% | 6% | 8% | 10% |
|---|---|---|---|---|---|---|---|---|---|---|
| CausE | 0.818 | 0.818 | 0.819 | 0.819 | 0.819 | 0.579 | 0.580 | 0.584 | 0.586 | 0.587 |
| KD-Label | 0.815 | 0.815 | 0.815 | 0.816 | 0.816 | 0.582 | 0.584 | 0.588 | 0.588 | 0.589 |
| AutoDebias | 0.810 | 0.815 | 0.818 | 0.826 | 0.832 | 0.569 | 0.580 | 0.587 | 0.589 | 0.590 |
| Res-IPS (ours) | **0.845** | **0.848** | **0.850** | **0.850** | **0.852** | **0.595** | **0.596** | **0.602** | **0.602** | **0.603** |
| Res-DR (ours) | **0.850** | **0.851** | **0.854** | **0.855** | **0.855** | **0.592** | **0.593** | **0.597** | **0.605** | **0.606** |

datasets. KUAIREC is a fully exposed dataset containing 4,676,570 video watching ratio records from 1,411 users for 3,327 videos. The records less than two are binarized to 0 and other records to 1.

**Baselines.** In our experiment, we take the widely-used **Matrix Factorization (MF)** [22] as the base model. We compare our methods with the debiasing methods: **IPS** [41, 42], **DR** [39, 52], **RD-IPS** [8], **RD-DR** [8], and multi-task learning approaches: **ESMM** [34], **Multi-IPS** [59], **Multi-DR** [59], **ESCM$^2$-IPS** [44] and **ESCM$^2$-DR** [44]. We also compared the methods using both biased data and unbiased data: **CausE** [1], **KD-Label** [30], **KD-Feature** [30] and **AutoDebias** [4].

**Experimental Protocols and Details.** We adopt three widely-used evaluation metrics: AUC, Recall@K (R@K), and NDCG@K (N@K) for debiasing performance evaluation. We set K = 5 for COAT and YAHOO! R3, and K = 50 for KUAIREC. All the experiments are implemented on Pytorch with Adam as the optimizer. We tune learning rate in $\{0.0001, 0.0005, 0.001, 0.005, 0.01, 0.05\}$, weight decay in $\{0, 1e-6, \ldots, 1e-1, 1\}$. For our methods, we tune $\alpha$ in $\{0.1, 0.5, 1\}$, $\beta$ in $\{0.1, 0.5, 1, 5, 10\}$, and $\gamma$ in $\{0.001, 0.005, 0.01, 0.05, 0.1\}$. In addition, we randomly split 5% of unbiased data from the test set to train models for all methods that require unbiased data[5].

**Performance Comparison.** Table 1 shows the real-world debiasing performance for varying methods on three datasets. First, most debiasing methods outperform the base model, i.e., MF (bias), demonstrating the necessity for debiasing in the presence of selection bias. Second, methods using both biased and unbiased data outperform the methods using only one of them, which indicates that there exists some non-overlap information that can benefit for debiasing between both biased and

---

[5]For all experiments, we use Tesla T4 GPU as the computational resource.

Table 3: Ablation study on **residual networks** in Res-DR method, with AUC, NDCG@K and Recall@K as evaluation metrics. The best result is bolded and the second is underlined.

| | Training loss | | COAT | | | YAHOO! R3 | | | KUAIREC | | |
|---|---|---|---|---|---|---|---|---|---|---|---|
| Method | $\Delta\delta$ | $\Delta p$ | AUC | R@5 | N@5 | AUC | N@5 | R@5 | AUC | N@50 | R@50 |
| ESCM$^2$-DR | × | × | 0.760 | 0.553 | 0.568 | 0.715 | 0.566 | 0.722 | 0.827 | 0.569 | 0.830 |
| Res-DR w/o $\Delta p$ $\Delta\delta$ | × | × | 0.763 | 0.544 | 0.568 | 0.716 | 0.560 | 0.715 | 0.831 | 0.570 | 0.836 |
| Res-DR w/o $\Delta p$ | ✓ | × | 0.783 | 0.561 | 0.573 | 0.734 | 0.630 | 0.781 | 0.833 | 0.570 | 0.849 |
| Res-DR w/o $\Delta\delta$ | × | ✓ | 0.768 | 0.555 | 0.581 | 0.721 | 0.645 | 0.791 | 0.841 | 0.579 | 0.836 |
| Res-DR | ✓ | ✓ | **0.793** | **0.588** | **0.607** | **0.750** | **0.654** | **0.803** | **0.854** | **0.595** | **0.860** |

Table 4: Ablation study on **loss components** in Res-IPS and Res-DR methods, with AUC, NDCG@K and Recall@K as evaluation metrics. The best result is bolded and the second is underlined.

| | Training loss | | COAT | | | YAHOO! R3 | | | KUAIREC | | |
|---|---|---|---|---|---|---|---|---|---|---|---|
| Method | $\mathcal{L}_{\text{CVR}}^{\mathcal{U}}$ | $\mathcal{L}_{\text{CVR}}^{\mathcal{B}\&\mathcal{U}}$ | AUC | N@5 | R@5 | AUC | N@5 | R@5 | AUC | N@50 | R@50 |
| ESCM$^2$-IPS | × | × | 0.757 | 0.514 | 0.558 | 0.729 | 0.559 | 0.714 | 0.815 | 0.577 | 0.860 |
| Res-IPS-None | × | × | 0.755 | 0.522 | 0.546 | 0.722 | 0.552 | 0.707 | 0.825 | 0.580 | 0.853 |
| Res-IPS-U | ✓ | × | 0.770 | 0.562 | 0.570 | 0.718 | 0.587 | 0.741 | 0.833 | 0.583 | 0.849 |
| Res-IPS-B&U | × | ✓ | **0.784** | 0.573 | 0.592 | 0.756 | 0.635 | 0.778 | 0.845 | 0.592 | 0.880 |
| Res-IPS | ✓ | ✓ | 0.777 | **0.575** | **0.601** | **0.759** | **0.639** | **0.785** | **0.849** | **0.601** | **0.885** |
| ESCM$^2$-DR | × | × | 0.760 | 0.553 | 0.568 | 0.715 | 0.566 | 0.722 | 0.827 | 0.569 | 0.830 |
| Res-DR-None | × | × | 0.765 | 0.544 | 0.550 | 0.714 | 0.575 | 0.735 | 0.824 | 0.562 | 0.823 |
| Res-DR-U | ✓ | × | 0.770 | 0.565 | 0.577 | 0.722 | 0.604 | 0.756 | 0.836 | 0.562 | 0.842 |
| Res-DR-B&U | × | ✓ | 0.790 | 0.574 | 0.601 | 0.744 | 0.640 | 0.782 | 0.848 | 0.586 | 0.848 |
| Res-DR | ✓ | ✓ | **0.793** | **0.588** | **0.607** | **0.750** | **0.654** | **0.803** | **0.854** | **0.595** | **0.860** |

unbiased data, and highlights the importance of leveraging both kinds of data. Meanwhile, the direct use of multi-task learning approaches to IPS and DR estimators cannot benefit the debiasing performance under hidden confounding. The proposed Res-IPS and Res-DR methods stably outperform previous methods on all three datasets, which provides empirical evidence of the existence of hidden confounding in the real-world recommendations, as well as the effectiveness of our methods for removing hidden confounding. Table 2 shows the results of AUC and NDCG@50 on KUAIREC with varying unbiased data ratios. The performance of all methods improves with increasing unbiased data ratio, and our method stably outperforms the baseline methods by a large margin.

**Ablation Studies.** The two losses $\mathcal{L}_{\text{CVR}}^{\mathcal{U}}$ and $\mathcal{L}_{\text{CVR}}^{\mathcal{B}\&\mathcal{U}}$ as well as the two residual terms $\Delta\delta$ and $\Delta p$ are crucial in the proposed multi-task learning approach. We further conduct ablation studies with respect to the residual networks and the training loss components, respectively. From Table 3, Res-DR using either the propensity residual network or the imputation residual network can stably outperform ESCM$^2$-DR, and Res-DR achieves the best performance when two residual networks are adopted together. When both propensity and imputation residual networks are removed, although Res-DR includes $\mathcal{L}_{\text{CVR}}^{\mathcal{U}}$ and $\mathcal{L}_{\text{CVR}}^{\mathcal{B}\&\mathcal{U}}$ losses, it has similar performance to ESCM$^2$-DR, which further indicates that the performance improvement of Res-DR can be attributed to the effectiveness of the residual networks. From Table 4, both Res-IPS and Res-DR methods without $\mathcal{L}_{\text{CVR}}^{\mathcal{U}}$ loss or without $\mathcal{L}_{\text{CVR}}^{\mathcal{B}\&\mathcal{U}}$ loss outperform ESCM$^2$-IPS and ESCM$^2$-DR. Similarly, our methods achieve the best performance when both two losses are preserved. Note that the model with $\mathcal{L}_{\text{CVR}}^{\mathcal{B}\&\mathcal{U}}$ loss performs better than the model with $\mathcal{L}_{\text{CVR}}^{\mathcal{U}}$ loss, which is attributed to $\mathcal{L}_{\text{CVR}}^{\mathcal{B}\&\mathcal{U}}$ corrects for biased learned propensities and biased imputed errors under the presence of hidden confounding. In addition, unbiased data does not significantly improve the debiasing performance when it is directly used to train the prediction model through $\mathcal{L}_{\text{CVR}}^{\mathcal{U}}$, due to the overfitting problem caused by the limited unbiased data size, which is consistent to the poor performance of MF (uniform) in Table 1. Meanwhile, minimizing $\mathcal{L}_{\text{CVR}}^{\mathcal{U}}$ does not provide any residual information for biased learned propensities and biased imputed errors.

**In-Depth Analysis.** The proposed methods contain the prediction model, imputation model, propensity model, and residual models, thus it is meaningful to investigate the effect of different optimization algorithms among these models on the debiasing performance. Specifically, we implement the following learning approaches on the Res-DR: (1) Joint Learning (JL) [52], which joint optimizes the prediction model and the imputation model. (2) Double Learning (DL) [11], which adds a parameter

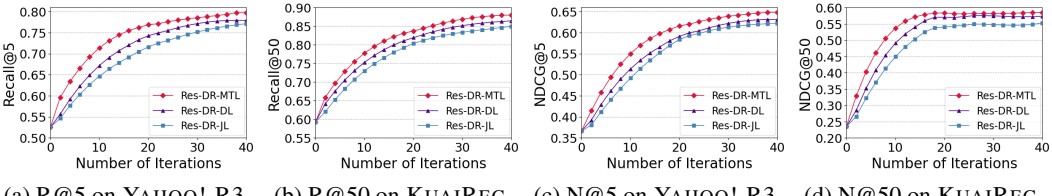

| (a) R@5 on YAHOO! R3 | (b) R@50 on KUAIREC | (c) N@5 on YAHOO! R3 | (d) N@50 on KUAIREC |

Figure 2: Joint learning (JL), double learning (DL), and multi-task learning (MTL) on Res-DR.

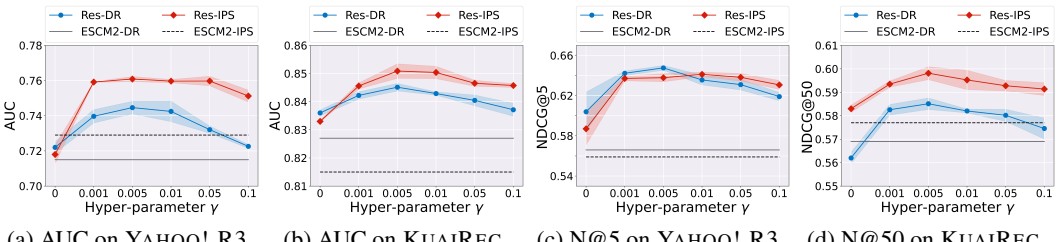

| (a) AUC on YAHOO! R3 | (b) AUC on KUAIREC | (c) N@5 on YAHOO! R3 | (d) N@50 on KUAIREC |

Figure 3: Effects of varying weights $\gamma$ of the consistency loss $\mathcal{L}_{\mathrm{CVR}}^{\mathcal{B}\&\mathcal{U}}$ on Res-IPS and Res-DR.

sharing mechanism between the prediction model and the imputation model based on JL. (3) Multi-Task Learning (MTL) adopted in our methods. Figure 2 shows the experiment results. Remarkably, MTL significantly outperforms JL and DL, whereas in Table 1 DR and MTL-based ESCM$^2$-DR perform similarly. This is because DR only has three training models (namely the prediction model, imputation model, and propensity model), whereas the proposed Res-DR has two additional residual models. As the number of models increases, JL can no longer bridge the models efficiently, which leads to a slow convergence and limited performance. Instead, DL increases the connection between models via timely parameter sharing. Finally, MTL trade-offs the different tasks in Res-DR to train all the models, leading to the optimal debiasing performance.

**Sensitivity Analysis.** We perform the sensitivity analysis of $\gamma$ on Res-IPS and Res-DR, as shown in Figure 3. Our methods achieve the optimal performance when $\gamma$ is moderate (0.005-0.01). This is because when $\gamma$ is too large, it hurts the performance of other tasks (e.g., CVR model training), and when $\gamma$ is too small, it makes the consistency loss be paid with less attention, so that the hidden confounding cannot be effectively removed. Res-IPS and Res-DR stably outperform ESCM$^2$-IPS and ESCM$^2$-DR under varying $\gamma$. This further illustrates the effectiveness the consistency loss.

## 6 Conclusion

This paper investigates the use of a few unbiased ratings to calibrate the learned propensities and imputed errors for removing hidden confounding. First, we theoretically reveal the biasedness of previous debiasing methods in the presence of hidden confounding. Next, we propose a multi-task debiasing residual networks learning approach for training the debiased prediction model. By building residual networks and calibrating the biased learned propensities and biased imputed errors, the prediction model is trained on both the calibrated IPS or DR losses and the unbiased dataset to ensure the unbiasedness. Extensive experiments on two benchmark datasets and a large-scale industrial dataset validate the effectiveness of our proposal. A limitation of this work is the use of slightly more model parameters due to the need to address hidden confounding with residual networks.

## Acknowledgements

This work is supported by National Key R&D Program of China (No. 2020YFE0204200) and National Natural Science Foundation of China (No. 62272437 and 61973329). Peng Wu is partially supported by the National Natural Science Foundation of China (No. 12301370) and the Disciplinary Funding of Beijing Technology and Business University (No. STKY202301). Zhi Geng is partially supported by the National Nature Science Foundation of China (No. 12071015).

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
