# OpenReview forum: "Removing Hidden Confounding in Recommendation: A Unified Multi-Task Learning Approach"
_NeurIPS.cc/2023/Conference — NeurIPS 2023 poster_

### Official Review · Reviewer_KtZ4 · 2023-06-19

**Soundness:** 3 good
**Presentation:** 3 good
**Contribution:** 2 fair
**Rating:** 4
**Confidence:** 4

**Summary:**

The paper addresses that there are hidden confounders in recommendations and existing methods cannot handle them well. \
The paper proposes a unified multi-task learning approach to tackle that problem. \
Specifically, they devise a residual network to calibrate the propensity and the imputed error by using unbiased data.


**Strengths:**

1. The paper is well-organized and presented.

2. Each loss function is technically solid.
- The proposed method is end-to-end multi-task learning with a debiasing residual network that simultaneously deals with selection bias, data sparsity, and hidden confounding.

3. The proposed method is validated through experiments.
- They adopt three real-world datasets, three metrics, and extensive baselines.


**Weaknesses:**

1. Motivation is weak.
- Hidden confounders are assumed to exist. There is no theoretical or experimental evidence for that.
- Proofs (Proposition 1, Theorem 2) are naive. They are proved on the assumption that is already the conclusion of the proofs.

2. Proposed end-to-end framework is not novel.
- There is already an end-to-end multi-task learning framework [7] removing hidden confounders with biased and unbiased data.
- Loss functions are from the existing methods. Most of them are already adopted in a multi-task manner in ESCM^2 [33].
- The unique contribution of this paper relies on unbiased data, which is hard to obtain in real-world applications.

3. Missing related work
- Balancing Unobserved Confounding with a Few Unbiased Rating in Debiased Recommendations, WWW 2023.
- StableDR: Stabilized Doubly Robust Learning for Recommendation on Data Missing Not At Random. ICLR 2023.
- Multiple Robust Learning for Recommendation. AAAI 2023.
- This work seems to tackle the same problem as the proposed work and needs to be included in the manuscript.

4. Notations are confusing
- There are no explicit loss forms. For example, what is $L^{B}_{CVR}$?
- The argument of loss functions is not organized. I cannot distinguish which is the training parameter and which is the fixed parameter.

**Questions:**

Please refer to Weaknesses.

**Limitations:**

There are many hyperparameters for the loss functions.

---

> ### Author Rebuttal · Authors · 2023-08-09
>
> We sincerely thank you for the helpful suggestions. **Below, we hope to address your concerns and questions to improve the clarity and quality of our paper.**
>
> > **W1:** Motivation is weak.
>
> - Hidden confounders are assumed to exist. There is no theoretical or experimental evidence for that.
>
> **Response to W1:** We thank the reviewer for pointing out this issue. Theoretically, the assumption of "unconfoundedness", i.e., the conditional independence of treatment and outcome given the covariates, cannot be tested with observational data, and thus we cannot theoretically prove the existence of hidden confounders [1]. However, experimentally, as summarized in a recent survey paper [2] on causal recommendation, a large amount of empirical evidence for the existence of hidden confounders has been explored.
>
> - Proofs (Proposition 1, Theorem 2) are naive. They are proved on the assumption that is already the conclusion of the proofs.
>
> **Response to W1:** We agree with the reviewer that it is not hard to derive Proposition 1 and Theorem 2, thus it should not be regarded as the core contribution of this paper. However, it provides us a very intuitive way to understand the motivation of calibrating the learned nominal propensities and nominal error imputations by a multi-task learning approach using unbiased data.
>
> > **W2:** Proposed end-to-end framework is not novel.
>
> - There is already an end-to-end multi-task learning framework [7] removing hidden confounders with biased and unbiased data.
>
> **Response to W2:** The reviewer raises an interesting concern. However, we have reflected this comment in Section 3.3, where we discuss the difference between our work and [7]. In summarize, [7] proposed to adopt sensitivity analysis from the causal inference literature to minimize the worst-case prediction loss. Whereas our work propose to tackle hidden confounding by a multi-task learning approach using unbiased data to calibrate learned nominal propensities and nominal error imputations.
>
> - Loss functions are from the existing methods. Most of them are already adopted in a multi-task manner in ESCM^2 [33].
>
> **Response to W2:** We thank the reviewer for pointing out this issue. However, we would like to emphasize that (a) the motivation and problem set are different: the main purpose of our work is to address unobserved confounding using unbiased data, whereas ESCM^2 only uses biased data to address selection bias; (b) the model setup is different: our work novelly introduces a residual imputation model and a residual propensity model to calibrate the learned biased nominal propensities and biased nominal error imputations due to unobserved confounding.
>
> - The unique contribution of this paper relies on unbiased data, which is hard to obtain in real-world applications.
>
> **Response to W2:** Thank you for the comment. We note that much recent work proposes to use a small amount of unbiased ratings to address selection bias, but fails to address unobserved confounding [3, 4]. Empirically, we add experiments in a supplemental one-page pdf that verify that even if the unbiased scoring has much much smaller scale (<0.1\%), the proposed method still has promising performance. To gather unbiased ratings, we may ask users to rate randomly selected items. This way, the propensities of observing different ratings are the same and the observed ratings are thus unbiased.
>
>
> > **W3:** Missing related work.
>
> - Balancing Unobserved Confounding with a Few Unbiased Rating in Debiased Recommendations, WWW 2023.
>
> **Response to W3:** First, we would like to kindly remind the reviewers that the work mentioned was published online in 30 April 2023, see: https://dl.acm.org/doi/abs/10.1145/3543507.3583495, while our submission was in May, which is in line with NeurIPS submission policy for baseline comparisons (due to the very close time). **We add comparisons that use this work as a baseline, and the experimental results validate the superiority of our proposal (please kindly refer to the one-page supplementary pdf for the detailed experimental results).**
>
> - StableDR: Stabilized Doubly Robust Learning for Recommendation on Data Missing Not At Random. ICLR 2023.
>
> - Multiple Robust Learning for Recommendation. AAAI 2023.
>
> **Response to W3:** We thank the reviewers for the useful suggestions. We kindly remind the reviewer that though both of them aim to combat the selection bias, neither of them tacking the effect of unobserved confounding. And as suggested by the reviewer, we will include them in our manuscript. **We also add comparisons that use these works as baselines in our one-page supplementary pdf.**
>
> > **W4:** Notations are confusing.
>
> - There are no explicit loss forms. For example, what is $L_{CVR}^B$?
>
> **Response to W4:** Please kindly refer to line 108.5 in our original manuscript, $L_{CVR}^B$ is either $L_{IPS}$ or $L_{DR}$.
>
> - The argument of loss functions is not organized. I cannot distinguish which is the training parameter and which is the fixed parameter.
>
> **Response to W4:** The reviewer raises an interesting concern. Following ESCM^2, our work uses multi-task learning to simultaneously train the propensity model, the CVR model, and the imputation model.
>
> ***
> **We hope the above discussion will fully address your concerns about our work, and we would really appreciate it if you could be generous in raising your score.** We look forward to your insightful and constructive responses to further help us improve the quality of our work. Thank you!
> ***
> **References**
>
> [1] Imbens, Guido W., and Donald B. Rubin. Causal inference in statistics, social, and biomedical sciences. 2015.
>
> [2] Luo, Huishi, et al. "A Survey on Causal Inference for Recommendation." 2023.
>
> [3] Wang, Xiaojie, et al. "Combating selection biases in recommender systems with a few unbiased ratings." WSDM, 2021.
>
> [4] Chen, Jiawei, et al. "AutoDebias: Learning to debias for recommendation." SIGIR, 2021.

---

> > ### Comment · Reviewer_KtZ4 · 2023-08-14
> > **Review response**
> >
> > Thank you for your response! It really helps me understand the manuscript.
> >
> > After I read your responses, I decided to raise my score from 3 to 4.
> >
> > Still, I cannot find any unique contribution of this paper, so I did not raise the score to the acceptance side.
> >
> > In my humble opinion, the contribution is somewhat incremental.

---

> > > ### Author Response · Authors · 2023-08-16
> > > **Thank you for your constructive comments and raising the score!**
> > >
> > > We are glad to know that many concerns in your original comments have been effectively addressed. We are very grateful for your constructive comments and questions, which helped improve the clarity and quality of our paper. We will provide more clarifications and explanations in the revised version. Thanks again!

---

### Official Review · Reviewer_csfC · 2023-07-03

**Soundness:** 3 good
**Presentation:** 3 good
**Contribution:** 3 good
**Rating:** 5
**Confidence:** 4

**Summary:**

This paper presents a critical examination of prevalent debiasing methods in recommendation systems and their limitations in addressing hidden confounding factors. The authors underline that current methods—propensity-based, multi-task learning, and bi-level optimization—fail to mitigate selection bias when unobserved confounding variables exist. In response, they propose a unified multi-task learning approach that incorporates a small set of unbiased ratings to calibrate nominal propensities and error imputations, thereby reducing the influence of hidden confounding factors. The approach utilizes a newly introduced consistency loss for calibration, advancing the field's theoretical understanding of bias and confounding in recommendation systems. The paper further validates its method through comprehensive experiments on three public benchmark datasets, including a large-scale industrial dataset. The outcomes affirm the efficacy of their approach in countering selection bias and hidden confounding. The study hence significantly contributes to the realm of recommender systems, revealing theoretical shortcomings in current debiasing approaches, and offering a novel, robust method for reducing bias in practice.

**Strengths:**

1. Proposed methods show quite significant improvements over the strong baselines.
2. Good to interpret each term in the equations. Very helpful for readers to intuitively understand the equations.
3. Code is attached, to be published with the paper.
4. Extended and complete experiments conducted on real-world datasets including a large-scale industrial dataset with many baselines and ablation studies.

**Weaknesses:**

1. Reference numbers for equations are missing. Please add the reference numbers to make referring more easily.
2. Where is Theorem 1? Or why not Lemma 1, Proposition 2, and Theorem 3?
3. Please refer to the proofs in the Appendix in the main text when the corresponding theorem or lemma are given.
4. Equation of L_{CVR}^{B&U} in Sect 4.4 is not very well-defined. The cross-entropy loss \delta(., .) is defined for binary random variables, however, neither L_{CVR}^B or L_{CVR}^U is a binary random variable. Please spell out the specific definition, as it’s also one of the main contribution.
5. Statistical significance analysis is missing.

**Questions:**

1. How good will the method be useful on real-world products? Assuming it is possible to get unbiased labels from evaluators, but usually at a much much smaller scale (<0.1%), and sparser (no consistent history of a user).

**Limitations:**

1. What are the assumptions about the hidden confounding? The main new term introduced L_{CVR}^{B&U} is not sufficiently discussed. If I understand correctly, the main assumptions are that the unbiased set has no hidden confounding and aligning the distribution of prediction on the biased set to that of the unbiased set should calibrate / de-confound the predictions.

---

> ### Author Rebuttal · Authors · 2023-08-09
>
> We sincerely thank you for the helpful suggestions. **Below, we hope to address your concerns and questions to improve the clarity and quality of our paper.** Below we categorized the reviewers' concerns into **Methodology**, **Experiments**, and **Clarity**.
>
> ### **Methodology**
> > **W4:** Equation of L_{CVR}^{B&U} in Sect 4.4 is not very well-defined. The cross-entropy loss \delta(., .) is defined for binary random variables, however, neither L_{CVR}^B or L_{CVR}^U is a binary random variable. Please spell out the specific definition, as it’s also one of the main contribution.
>
> **Response to W4:** We apologize for the lack of clarity and its caused misunderstanding by the reviewer. **In fact, we use \delta(. , .) to denote generic loss functions** in our manuscript and **are not limited to cross-entropy losses**. The reviewer is correct that **for L_{CVR}^{B&U}, we use the square of the difference between L_{CVR}^B and L_{CVR}^U.** This can also be verified in our released codes in the supplementary material. Instead, when defining the ideal loss, since this work considers binary ratings, we use cross-entropy loss in our experiments. We will clarify it in our revised version.
>
> > **Limitations:** What are the assumptions about the hidden confounding? The main new term introduced L_{CVR}^{B&U} is not sufficiently discussed. If I understand correctly, the main assumptions are that the unbiased set has no hidden confounding and aligning the distribution of prediction on the biased set to that of the unbiased set should calibrate / de-confound the predictions.
>
> **Response to Limitations:** With respect to the assumptions about hidden confounding, **the reviewer is correct: the main assumptions are that the unbiased set has no hidden confounding.**  Nevertheless, we would like to clarify that this is a truth for unbiased datasets, instead of an assumption about hidden confounding [1, 2]. To gather unbiased ratings, we may ask users to rate randomly selected
> items. This way, the propensities of observing different ratings are the same and the observed ratings are thus unbiased.
>
> For the term L_{CVR}^{B&U}, **the reviewer is also correct: we align the distribution of prediction on the biased set to that of the unbiased set to calibrate / de-confound the predictions.** The core idea to tackle the hidden confounding by a multi-task learning approach is **the using of unbiased data to calibrate learned nominal propensities and nominal error imputations,** which motivates our design for the consistency loss L_{CVR}^{B&U}.
>
> ### **Experiments**
>
> > **W4:** Statistical significance analysis is missing.
>
> **Response to W4:** As suggested by the reviewer, we add the statistical significance test in the attachment PDF. The result shows that Res-IPS and Res-DR outperform other baseline methods.
>
> > **Questions:** How good will the method be useful on real-world products? Assuming it is possible to get unbiased labels from evaluators, but usually at a much much smaller scale (<0.1%), and sparser (no consistent history of a user).
>
> **Response to Questions:** Thanks for the question. Actually, we conduct the unbiased data ratio experiments from **2% to 10%** in the submitted paper. As suggested by the reviewer, we add a unbiased data ratio experiments from **0.05% to 10% with more subtle step size** in the attachment PDF. It shows that some of the baseline methods (like autodebias on KuaiRec) performance can drop off dramatically with very little unbias data. However, Res-IPS and Res-DR still outperform the baseline methods with varying unbiased data ratio. Please kindly refer to the attachment PDF for more details.
>
>
> ### **Clarity**
> > **W1:** Reference numbers for equations are missing. Please add the reference numbers to make referring more easily.
>
> **Response to W1:** Thanks for your helpful suggestion, and we will add reference numbers for equations in our revised manuscript.
>
> > **W2:** Where is Theorem 1? Or why not Lemma 1, Proposition 2, and Theorem 3?
>
> **Response to W2:** We thank the reviewer for pointing out this issue, it seems our template generates the wrong number of lemmas, propositions, and theorems, and we apologize to this expected issue and will adjust the numbering of lemmas, propositions, and theorems in the revised version.
>
> > **W3:** Please refer to the proofs in the Appendix in the main text when the corresponding theorem or lemma are given.
>
> **Response to W3:** We agree with you and will add the "please refer to the proofs in the Appendix" in our main text when the corresponding theorem or lemma are given. Thanks again for helping us to improve the clarity of our paper.
>
> ***
> **We hope the above discussion will fully address your concerns about our work, and we would really appreciate it if you could be generous in raising your score.** We look forward to your insightful and constructive responses to further help us improve the quality of our work. Thank you!
> ***
> **References**
>
> [1] Wang, Xiaojie, et al. "Combating selection biases in recommender systems with a few unbiased ratings." Proceedings of the 14th ACM International Conference on Web Search and Data Mining. 2021.
>
> [2] Chen, Jiawei, et al. "AutoDebias: Learning to debias for recommendation." Proceedings of the 44th International ACM SIGIR Conference on Research and Development in Information Retrieval. 2021.

---

> > ### Comment · Reviewer_csfC · 2023-08-14
> >
> > Thank you for the detailed responses.
> >
> > Given the additional experimental results on statistical significance and ratio of unbiased data in the appended pdf, I will raise my rating to 5.

---

> > > ### Author Response · Authors · 2023-08-16
> > > **Thank you for your constructive comments and raising the score!**
> > >
> > > We are glad to know that your concerns have been effectively addressed. We are very grateful for your constructive comments and questions, which helped improve the clarity and quality of our paper. Thanks again!

---

### Official Review · Reviewer_J6YV · 2023-07-05

**Soundness:** 3 good
**Presentation:** 4 excellent
**Contribution:** 3 good
**Rating:** 6
**Confidence:** 4

**Summary:**

This paper highlights the prevalent issue of selection bias in recommender systems, emphasizing the often-overlooked aspect of hidden confounding. Existing approaches and their limitations are discussed, with a special focus on hidden confounders. The authors then introduce a unified multi-task learning approach which utilizes a small set of unbiased ratings to calibrate nominal propensities and error imputations. This approach aims to handle hidden confounding, thereby achieving unbiased learning.

**Strengths:**

- The paper does an excellent job of dissecting existing propensity-based and multi-task learning methods theoretically to demonstrate how they can lead to biased learning in the presence of hidden confounding.

- Although the multi-task learning approach for debiasing is not novel, the specific setting of calibrations on unbiased data is novel. It uniquely uses a few unbiased ratings to calibrate learned propensities and imputed errors from biased data, aiming to eliminate the biases caused by hidden confounding. The unified loss is also justified by theoretical analysis.

- The proposed methods are extensively validated on three publicly available benchmark datasets, including a large-scale industrial dataset. This broad empirical evidence strengthens the claims made in the paper.

**Weaknesses:**

- The theoretical result is weak. It states that if the consistency loss is zero then the calibrated loss is unbiased. While the result does help justify the design of the loss function, it does not provide any guarantee regarding the unbiasedness of the learned model. The consistency loss might not reach zero or even be small enough to reduce the bias of the calibrated loss.

- The overall model contains multiple loss components and multiple parameters. While the authors point out the potential issues of the existing minimax framework, the proposed solution requires more parameter tuning.



**Questions:**

In Figure 2, the higher weight of the consistency loss actually reduces the performance of the model which is not very intuitive. Based on the propositions, reduced consistency loss leads to unbiased calibration loss. With higher weight, we expect smaller consistency loss and in general a better calibration loss. Why do we observe a regression in the AUC?

---

> ### Author Rebuttal · Authors · 2023-08-09
>
> We sincerely thank you for the helpful suggestions. **Below, we hope to address your concerns and questions to improve the clarity and quality of our paper.**
>
> > **W1:** The theoretical result is weak. It states that if the consistency loss is zero then the calibrated loss is unbiased. While the result does help justify the design of the loss function, it does not provide any guarantee regarding the unbiasedness of the learned model. The consistency loss might not reach zero or even be small enough to reduce the bias of the calibrated loss.
>
> **Response to W1:** Thanks for your comments.  We are glad that the reviewer agree that the theoretical results do help justify the design of the loss function. Actually, all the theoretical results are intended to provide us with a very intuitive way to understand the design of each component of the proposed method, which is important to make the presentation clearer.
>
> We fully agree with the reviewer that the consistency loss might not reach zero or even be small enough to reduce the bias of the calibrated loss. However, we would like to clarify that **this is a general problem with multi-tasking learning and is not unique to our method**, see references [1, 2, 3].  Actually, **all the methods cannot guarantee the exact zero for the loss functions.** For example,  propensity-based methods are unbiased when the propensity model is correctly specified, but no methods can guarantee and verify they obtain 100\% accurate estimation of propensities.
>
> In summary, the proposed method provides a framework to combat the risk of hidden/unmeasured confounding. In particular, **the consistency loss provides a ideal and natural optimization direction to debias in the presence of hidden/unmeasured confounding, and with theoretical guarantees when its value is zero.**
>
>
> > **W2:** The overall model contains multiple loss components and multiple parameters. While the authors point out the potential issues of the existing minimax framework, the proposed solution requires more parameter tuning.
>
> **Response to W2:**  Thanks for your comments. Indeed, we agree with the reviewer that the proposed methods requires more parameter tuning. However, we would like to clarity that due to the usage of the unbiased dataset and the carefully designed model structure in the proposed method, it is easy to obtain several hyper-parameter combination of $\alpha, \beta, \gamma$ with promising performance in practice.
>
> Actually, the proposed method mainly adds only an additional hyper-parameter $\gamma$ empirically. This is because we can first implement ESCM$^2$ method (the state-of-the-art multi-task learning method for combating the selection bias) and then take it as a pre-trained model. Thus, **we essentially have only one hyper-parameter ($\gamma$) instead of three, which is easy to implement (Please kindly refer to our codes in the Supplementary Material).**
>
> > **Q1:** In Figure 2, the higher weight of the consistency loss actually reduces the performance of the model which is not very intuitive. Based on the propositions, reduced consistency loss leads to unbiased calibration loss. With higher weight, we expect smaller consistency loss and in general a better calibration loss. Why do we observe a regression in the AUC?
>
> **Response to Q1:**  We thank the reviewer for pointing out this issue and apologize for the lack of clarity. As shown in Figure 2, our method performs best when $\gamma$ is moderate. Here are the reasons.
>
> - **When $\gamma$ is too large**, **though a higher weight can mitigate the shift between training set and testing set caused by unobserved confounding, it hurts the performance of other tasks** according the loss function $L_{Res}$ in lines 179-180. This is becase much attention is paid to $L_{CVR}^{B\\&U}$ (which strictly relies on $L_{CVR}^{B}$ and $L_{CVR}^{U}$), whereas the other loss functions (including $L_{CTR}$, $L_{IMP}$, $L_{CVR}^{B}$, $L_{CTCVR}^{B}$, and $L_{CVR}^{U}$) cannot be sufficiently optimized, which prevents the debiased learning of the CVR model (via $L_{CVR}^{B}$, $L_{CTCVR}^{B}$, and $L_{CVR}^{U}$), accurate learned propensities (via $L_{CTR}$), and accurate imputed errors (via $L_{IMP}$) from being realized, leading to sub-optimal prediction performance.
>
> -  **When $\gamma$ is too small**, it makes this balance loss $L_{CVR}^{B\\&U}$ be paid with less attention, so that **the residuals are no longer sufficiently updated to combat the unobserved confounding.**
>
> Therefore, our methods perform best when $\gamma$ is moderate.
>
> ***
>
> **We sincerely thank you for your feedback and will provide more clarifications and explanations in the revised version, and welcome any further technical advice or questions on this work and we will make our best to address your concerns.**
>
> ___
> **References**
>
> [1] Ma, Xiao, Liqin Zhao, Guan Huang, Zhi Wang, Zelin Hu, Xiaoqiang Zhu, and Kun Gai. Entire space multi-task model: An effective approach for estimating post-click conversion rate. In SIGIR, 2018.
>
> [2] Zhang, Wenhao, Wentian Bao, Xiao-Yang Liu, Keping Yang, Quan Lin, Hong Wen, and Ramin Ramezani. Large-scale causal approaches to debiasing post-click conversion rate estimation with multi-task learning. In WWW, 2020.
>
> [3] Wang, Hao, Tai-Wei Chang, Tianqiao Liu, Jianmin Huang, Zhichao Chen, Chao Yu, Ruopeng Li, and Wei Chu. ESCM2: Entire Space Counterfactual Multi-Task Model for Post-Click Conversion Rate Estimation. In SIGIR, 2022.

---

### Official Review · Reviewer_Uwjf · 2023-07-07

**Soundness:** 3 good
**Presentation:** 3 good
**Contribution:** 3 good
**Rating:** 6
**Confidence:** 3

**Summary:**

This paper studies unbiased learning in recommendation systems in the presence of hidden confounding. The authors first theoretically analyze the limitations of previous MTL methods and those combine some unbiased data and then design a unified MTL debiasing method by calibrating the learned nominal propensities and error imputations using a novel consistency loss. Extensive experiments on benchmark datasets have shown the effectiveness of their proposed method.



**Strengths:**

1. The work is well-motivated with the analysis of the limitations of existing work in the presence of hidden confounding.
2. The presentation of the method design is clear and vivid. Each part of the optimization goal is stated in a good structure.
3. Extensive experiments of comparison with various baselines and in-depth analysis are persuasive, which shows the effectiveness of the designed method and the function of each component.

**Weaknesses:**

The novelty of this work lies in the proposed consistency loss that utilizes unbiased data to calibrate the learned nominal propensities and imputed errors from the biased data. However, the theoretical analysis of this term seems trivial (Proposition 3). Maybe the trade-off between different terms in $\mathcal{L}_{Res}$ can be discussed.


**Questions:**

How do you tune the hyperparameters $\alpha, \beta, \gamma$? Do you use a validation set? If so, how do other baselines use the validation set?

**Limitations:**

The authors have a discussion of the limitations in the Conclusion section.

---

> ### Author Rebuttal · Authors · 2023-08-09
>
> We sincerely thank you for the helpful suggestions. **Below, we hope to address your concerns and questions to improve the clarity and quality of our paper.**
>
> > **W1:** The novelty of this work lies in the proposed consistency loss that utilizes unbiased data to calibrate the learned nominal propensities and imputed errors from the biased data. However, the theoretical analysis of this term $\mathcal{L}_{res}$ seems trivial (Proposition 3).
>
> **Response to W1:** We agree with the reviewer that it is not hard to derive Proposition 3, thus it should not be regarded as the core contribution of this paper. However, **it provides us a very intuitive way to understand the design of $\mathcal{L}_{CVR}^{B\\&U}$, which is important to make the presentation clearer.**
>
> > **W2:**  Maybe the trade-off between different terms in $\mathcal{L}_{res}$  can be discussed.
>
> **Response to W2:** We thank the reviewer for pointing out this issue. Please kindly refer to the discussion of the trade-off between different terms in $\mathcal{L}_{res}$ in below.
>
> - First, **we perform the sensitivity analysis of $\gamma$, which is the weight for $\mathcal{L}_{CVR}^{B\\&U}$ in Res-IPS and Res-DR.**  The associated results are displayed in Figure 3, which indicates that our methods perform best when $\gamma$ is moderate.
> - This is because when $\gamma$ is too large, it hurts the performance of other tasks (e.g., debiased CVR model training), and when $\gamma$ is too small, it makes this balance loss be paid with less attention, so that the residuals are no longer sufficiently updated.
> - Meanwhile, **we conduct ablation studies for Res-IPS and Res-DR, with respect to the residual components and the training losses, respectively.** The results are shown in Table 3 and Table 4, from which one can see that our methods reach the best performance when both two losses are preserved.
>
> > **Q1:** How do you tune the hyperparameters $\alpha, \beta, \gamma$?  **Q2:** Do you use a validation set? If so, how do other baselines use the validation set?
>
> **Response to Q1 and Q2:**  We thank the reviewer for raising this question. **Yes, we use a validation set to tune the hyper-parameters** and to decide the stop criterion **for all baseline methods.** Specifically, **for the stop criterion,** the training process is finished **when the predicted AUC value on the validation set no longer increases.** Also, **for tuning the hyper-parameters,** we tune $\alpha$ in $\\{0.1, 0.5, 1\\}$, $\beta$ in $\\{0.1, 0.5, 1, 5, 10\\}$, and $\gamma$ in $\\{0.001, 0.005, 0.01, 0.05, 0.1\\}$ and use **grid search** to choose the hyper-parameters **which can achieve the highest AUC value on the validation set.**
>
> ***
>
> **We sincerely thank you for your feedback and will provide more clarifications and explanations in the revised version, and welcome any further technical advice or questions on this work and we will make our best to address your concerns.**

---

### Official Review · Reviewer_y77Q · 2023-07-15

**Soundness:** 2 fair
**Presentation:** 2 fair
**Contribution:** 3 good
**Rating:** 6
**Confidence:** 3

**Summary:**

The paper shows that existing approaches that are based on multi-task learning or take advantage of unbiased data have theoretical limitations in the problem of recommendation debiasing when hidden confounding is present. The paper proposes to address these limitations by a unified multi-task learning approach whose key idea is to remove hidden confounding by calibrating learned nominal propensities and nominal error imputations by a consistency loss on unbiased data. The paper conducts extensive experiments to confirm the effectiveness of the proposed approach on 3 widely used recommendation datasets.

**Strengths:**

S1: The idea of tackling hidden confounding by a multi-task learning approach using unbiased data to calibrate learned nominal propensities and nominal error imputations is novel to me.

S2: The paper compares the proposed approach against a wide range of existing representative approaches on 3 benchmark recommendation datasets under commonly used evaluation metrics.

S3: The paper does a great job in reviewing major research directions in the problem of recommendation debiasing and the references cover a full spectrum of related works.


**Weaknesses:**

W1: It is not fair to claim that the paper is the first to "perform theoretical analysis to reveal the possible failure of previous approaches" as Ding et al. has already done a majority of such theoretical analysis [7], which is cited by the paper.

W2: The proofs for the theoretical results presented in the paper are a bit hand waved. Take Theorem 2 as an example. It is not unclear what is the space spanned by x and why the fact that \bar{p} will not degenerate a point in the space spanned by x leads to the existence of a positive constant \eta that satisfies the condition.

[7] Addressing unmeasured confounder for recommendation with sensitivity analysis

**Questions:**

Q1: Do you have any explanations why the proposed approach underperforms RD-DR [7] in terms of R@50 on the KUAIREC dataset?

Q2: Is N@K in Tables 1 and 3 NDCG@K? If so, it is important to clarify the meaning of N@5 (and R@5) in the main text as it is not common to abbreviate NDCG@K as N@5.

[7] Addressing unmeasured confounder for recommendation with sensitivity analysis

---

> ### Author Rebuttal · Authors · 2023-08-09
>
> We sincerely thank you for the helpful suggestions. **Below, we hope to address your concerns and questions to improve the clarity and quality of our paper.**
>
>   > **W1:** It is not fair to claim that the paper is the first to "perform theoretical analysis to reveal the possible failure of previous approaches" as Ding et al. has already done a majority of such theoretical analysis [7], which is cited by the paper.
>
> **Response to W1:** The reviewer might have misunderstood. We do not claim that we are the first to do this. In lines 7-13, we simply say "We **first** perform theoretical analysis ...... **Then**, we propose ....".  : ) Please kindly note that we explicitly cite such theoretical analysis in [7] in line 101 and line 104. We fully agree that this part is not our theoretical contribution. Instead, this paper provides a novel proof to shown the biasedness of AutoDebias and LTD under unobserved confounding in line 122.
>
> Below, we kindly provide a detailed comparison table to illustrate the difference between [7] and our work for enhancing clarity of our work.
>
> |**Paper**      | **[7] Addressing unmeasured confounder for recommendation with sensitivity analysis** | **Ours** |
> |  :---        |    :----   |     :--- |
> |Dataset(s)| **Only** biased data| Biased data **with a few unbiased data**|
> |Method|**Sensitivity analysis,** which is a **Minimax** approach   | Using **unbiased data** to **calibrate** learned nominal propensities and nominal error imputations |
> |Theoretical result |Only **mitigate the unobserved confounding**, see the last paragraph of section 3.2 in [7] :" **Instead of aiming to eliminate the unmeasured confounding thoroughly,** the proposed RD framework ..., which provides a flexible way to **mitigate the unmeasured confounding**"| **Entirely eliminate the unobserved confounding** with a few unbiased ratings|
> |Assumption| **Assumption** that "assume the nominal propensity score can be expressed as ..." in Section 3.2 in [7] |**No additional assumptions,** except the need to use unbiased data|
> |Unbiasedness|**No**| **Yes (with the help of the unbiased data)**|
> |Learning Approach|**Joint learning,** i.e., alternative update (See Alg. 1 in [7])|**Multi-task learning**|
>
> > **W2:** The proofs for the theoretical results presented in the paper are a bit hand waved. Take Theorem 2 as an example. It is not unclear what is the space spanned by x and why the fact that \bar{p} will not degenerate a point in the space spanned by x leads to the existence of a positive constant \eta that satisfies the condition.
>
> **Response to W2:**  We apologize for the lack of clarity.  **Next, we give a more detailed proof of Theorem 2**.
>
> In the setting of MNAR, $r_{u,i}$ has a non-zero effect on $o_{u,i}$, so $p_{u,i} \neq \bar p_{u,i}$ according to the definition of MNAR.  Thus, for some $\epsilon>0$,   there exist  positive constants  $\delta_\epsilon,N_1(\epsilon)>0$,  such that for all $|\mathcal{D}|>N_1(\epsilon)$,
> $$  \mathbb{P}(  | \bar p_{u,i} -  p_{u,i} | \geq \epsilon )\geq \delta_\epsilon > 0.   $$
>   Without loss of generality, if $\hat p_{u,i}$ is the learned propensity by fitting $o_{u,i}$ with $x_{u,i}$, then $\hat p_{u,i}$ essentially estimates $\mathbb{E}[o_{u,i}=1|x_{u,i}] =\mathbb{P}(o_{u,i}=1 | x_{u,i}) = p_{u,i}$. Then, there exists some  $N_2(\epsilon)>0$,  such that   for all $|\mathcal{D}|>N_2(\epsilon)$,
> $$   \mathbb{P}(  | \hat p_{u,i} -  p_{u,i} | \geq \frac{ \epsilon}{2} )<\frac{\delta_\epsilon}{4}.  $$
>   Thus, if $|\mathcal{D}|> \max \\{ N_1(\epsilon), N_2(\epsilon) \\}$, we have
> $$
>   \mathbb{P}( | \bar p_{u,i} -  p_{u,i} | \geq \epsilon, | \hat p_{u,i} -  p_{u,i} | < \frac{ \epsilon}{2} )
> = \mathbb{P}(  | \bar p_{u,i} -  p_{u,i} | \geq \epsilon )-\mathbb{P}( | \bar p_{u,i} -  p_{u,i} | \geq \epsilon, | \hat p_{u,i} -  p_{u,i} |  \geq \frac{ \epsilon}{2} )
> \geq {\delta_\epsilon} -\frac{\delta_\epsilon}{4} =\frac{3}{4} \delta_\epsilon.
> $$
>   Let $\eta=\epsilon/2$. Since $\\{| \bar p_{u,i} -  p_{u,i} |\geq \epsilon, | \hat p_{u,i} -  p_{u,i} | < { \epsilon}/{2}\\} \subset \\{| \hat p_{u,i} - \bar p_{u,i} | \geq \eta\\}$, we have
> $$  \mathbb{P}(  | \hat p_{u,i} - \bar p_{u,i} | \geq \eta )\geq \mathbb{P}( | \bar p_{u,i} -  p_{u,i} |\geq \epsilon  ,| \hat p_{u,i} -  p_{u,i} | <\frac{ \epsilon}{2} )> \frac{3}{4}\delta_\epsilon.  $$
>   Thus,
> $$ \lim_{|\mathcal{D}| \to\infty} \mathbb{P}(  | \hat p_{u,i} - \bar p_{u,i} | \geq \eta) \geq\frac{3}{4}\delta_\epsilon>0.  $$
>   Similarly, it can be shown that
> $$ \lim_{|\mathcal{D}| \to\infty} \mathbb{P}(  | \hat \delta_{u,i} - \bar g_{u,i} | \geq \eta) >0.  $$
>
> > **Q1:** Do you have any explanations why the proposed approach underperforms RD-DR [7] in terms of R@50 on the KUAIREC dataset?
>
> **Response to Q1:**   Thanks for your comments. We would like to emphasize that the **overall performance** of the proposed methods is the best among all the methods on all three datasets. In addition, we do carefully tune the parameters of all baseline models, so that some of them will have a competitive result.
>
> > **Q2:** Is N@K in Tables 1 and 3 NDCG@K? If so, it is important to clarify the meaning of N@5 (and R@5) in the main text as it is not common to abbreviate NDCG@K as N@5.
>
> **Response to Q2:** We thank the reviewer for pointing out this issue. Yes, in Tables 1 and 3, N@K means NDCG@K and R@K means Recall@K. We will clarify this in our revised manuscript.
>
> ***
>
> **We sincerely thank you for your feedback and will provide more clarifications and explanations in the revised version, and welcome any further technical advice or questions on this work and we will make our best to address your concerns.**

---

> > ### Comment · Reviewer_y77Q · 2023-08-15
> >
> > Thanks for address my comments, which much improves the quality of the paper. I decided to raise my score to 6

---

> > > ### Author Response · Authors · 2023-08-16
> > > **Thank you for your constructive comments and raising the score!**
> > >
> > > We are glad to know that your concerns have been effectively addressed. We are very grateful for your constructive comments and questions, which helped improve the clarity and quality of our paper. Thanks again!

---

### Author Rebuttal · Authors · 2023-08-10

We sincerely thank you for all helpful suggestions. We add statistical significant results and more detailed results of the unbiased data ratio in the attached PDF. We welcome any further technical advice or questions on this work and we will make our best to address your concerns.

---

### Decision · Program_Chairs · 2023-09-21

**Decision:**

Accept (poster)

**Comment:**

All the reviewers agree that the paper studies a well-motivated and significant problem of debiased learning of recommender systems, and proposes an effective multi-task learning approach that is robust in the presence of hidden confounders. The experiments are extensive and the reviewers appreciated the authors' several clarifications in their responses. The authors performed additional sensitivity analysis experiments, analyzed the performance of their method as more unbiased data becomes available, and statistical significance results. All of these additions substantially strengthen the paper and should be included in the revision.